# CO-MANIFOLD LEARNING WITH MISSING DATA

## ABSTRACT

Representation learning is typically applied to only one mode of a data matrix, either its rows or columns. Yet in many applications, there is an underlying geometry to both the rows and the columns. We propose utilizing this coupled structure to perform co-manifold learning: uncovering the underlying geometry of both the rows and the columns of a given matrix, where we focus on a missing data setting. Our unsupervised approach consists of three components. We first solve a family of optimization problems to estimate a complete matrix at multiple scales of smoothness. We then use this collection of smooth matrix estimates to compute pairwise distances on the rows and columns based on a new multi-scale metric that implicitly introduces a coupling between the rows and the columns. Finally, we construct row and column representations from these multi-scale metrics. We demonstrate that our approach outperforms competing methods in both data visualization and clustering.

## 1 INTRODUCTION

Dimension reduction plays a key role in exploratory data analysis, data visualization, clustering and classification. Techniques range from the classical PCA and nonlinear manifold learning to deep autoencoders (Tenenbaum et al., 2000; Roweis & Saul, 2000; Belkin & Niyogi, 2003; Coifman & Lafon, 2006; Vincent et al., 2008; Rifai et al., 2011; Kingma & Welling, 2014). These techniques focus on only one mode of the data, often the observations (columns) which are measurements in a high-dimensional feature space (rows), and exploit correlations among the features to reduce the dimension of the feature vectors and obtain the underlying low-dimensional geometry of the observations. Yet for many data matrices, for example in gene expression studies, recommendation systems, sensor networks, and word-document analysis, correlations exist among both observations and features. In these cases, we seek a method that can exploit the correlations among both the rows and columns of a data matrix to better learn lower-dimensional representations of both. Biclustering methods, which extract *distinct* biclusters along both rows and columns, give a partial solution to performing simultaneous dimension reduction on the rows and columns of a data matrix. In certain settings, however, assuming a bi-clustering model is too restrictive and results in breaking up smooth geometries into artificial disjoint clusters that do not match the actual structure of the data. This can occur when the true geometry is one of overlapping rather than disjoint clusters, for example in word-document analysis (Ahn et al., 2010), or when the underlying structure is not one of clusters at all but rather a smooth manifold (Gavish & Coifman, 2012). Thus, we consider a more general viewpoint: data matrices possess geometric relationships between their rows (features) and columns (observations) such that both modes lie on low-dimensional *manifolds*. Furthermore, the relationships between the rows may be informed by the relationships between the columns, and vice versa. Several recent papers (Gavish & Coifman, 2012; Ankenman, 2014; Mishne et al., 2016; Shahid et al., 2016; Mishne et al., 2017; Yair et al., 2017) exploit this coupled relationship to co-organize matrices and infer underlying row and column embeddings.

Further complicating the story is that such matrices may suffer from missing values, due to measurement corruptions and limitations. Missing values can sabotage efforts to learn the low dimensional manifold underlying the data. Specifically, kernel-based methods rely on calculating a similarity matrix between observations, whose eigendecomposition yields a new embedding of the data. As the number of missing entries grows, the distances between points are increasingly distorted, resulting in poor representation of the data in the low-dimensional space (Gilbert & Sonthalia, 2018). Matrix completion algorithms assume the data is low-rank and fill in the missing values by fitting a global linear subspace to the data. Yet, this may fail when the data lies on a nonlinear manifold.

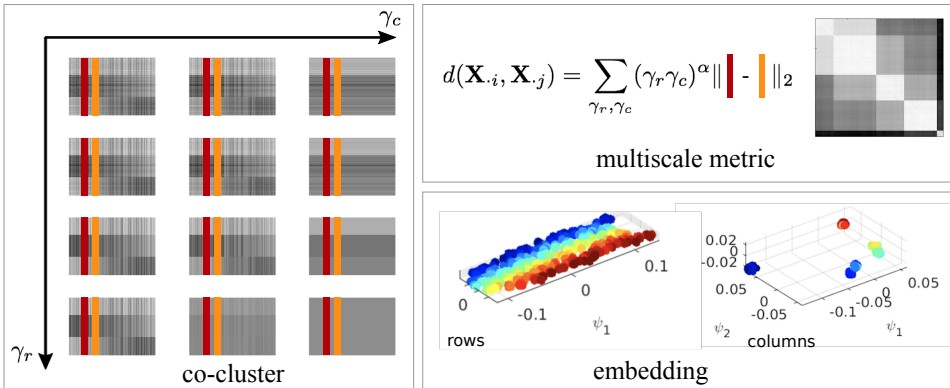

Figure 1: The three components of our approach: 1) smooth estimates of a matrix with missing entries at multiple scales via co-clustering, 2) a multi-scale metric using the smooth estimates across all scales, yielding an affinity kernel between rows/columns, and 3) nonlinear embeddings of the rows and columns. The multiscale metric between two columns (red and orange) is a weighted Euclidean distance between those columns at multiple scales, given by solving the co-clustering for increasing values of the cost parameters $\gamma_r$ and $\gamma_c$.

Manifold learning in the missing data scenario has been addressed by a few recent papers. Nonlinear Principle Component Analysis (NLPCA) (Scholz et al., 2005) uses an autoencoder neural network, where the middle layer serves to learn a low-dimensional embedding of the data, and the trained autoencoder is used to fill in missing values. Missing Data Recovery through Unsupervised Regression (Carreira-Perpin & Lu, 2011) first fills in the missing data with linear matrix completion methods, then calculates a non-linear embedding of the data and incorporates this embedding in an optimization problem to fill in the missing values. Recently Gilbert & Sonthalia (2018) proposed MR-MISSING which first calculates an initial distance matrix using only non-missing entries and then uses the increase-only-metric-repair (IOMR) method to fix the distance matrix so that it is a metric from which they calculate an embedding. None of these methods consider the co-manifold setting, where the coupled structure of the rows and the columns can be used to fill in the data, and to calculate an embedding.

In this paper, we introduce a new method for performing joint dimension reduction on the rows and columns of a data matrix, which we term co-manifold learning, in the missing data setting. We build on two recent lines of work on co-organizing the rows and columns of a data matrix (Gavish & Coifman, 2012; Mishne et al., 2016; 2017) and convex optimization methods for performing co-clustering (Chi et al., 2017; 2018). The former provide a flexible framework for jointly organizing rows and columns but lacks algorithmic convergence guarantees. The latter provides convergence guarantees but does not take full advantage of the multiple scales of the data revealed in the solution.

In the first stage of our approach, rather than inferring biclusters at a single scale, we use a multi-scale optimization framework to fill in the data at fine to coarse scales while imposing smoothness on both the rows and the columns. The scales of the solutions are encoded in a pair of joint cost parameters along the rows and columns. Next, we define a new multi-scale metric based on the filled-in matrix across all scales, which we then use to calculate nonlinear embeddings of the rows and columns. Thus our approach yields three results: a collection of smoothed estimates of the matrix, pairwise distances on the rows and columns that better estimate the geometry of the complete data matrix, and corresponding nonlinear embeddings (see Figure 1). We will demonstrate in experimental results that our method reveals meaningful representations in coupled datasets with missing entries, whereas other methods are capable of revealing a meaningful representation only along one of the modes.

The paper is organized as follows. We present the optimization framework in Section 2, the new multi-scale metric for co-manifold learning in Section 3 and experimental results in Section 4.

## 2 CO-CLUSTERING AN INCOMPLETE DATA MATRIX

We seek a collection of complete matrix approximations of a partially observed data matrix $\mathbf{X} \in \mathbb{R}^{m \times n}$ that have been smoothed along their row and columns to varying degrees. This collection will serve in computing row and column multi-scale distances to better estimate the pairwise distances of the complete data matrix. Let $[m]$ denote the set of indices $\{1, \ldots, m\}$, and let $\Theta \subseteq [m] \times [n]$ be a subset of the indices that correspond to observed entries of $\mathbf{X}$, and let $\mathcal{P}_\Theta$ denote the projection operator of $m \times n$ matrices onto an index set $\Theta$, i.e. $[P_\Theta(\mathbf{X})]_{ij}$ is $x_{ij}$ if $(i, j) \in \Theta$ and is 0 otherwise. We seek a minimizer $\mathbf{U}(\gamma_r, \gamma_c)$ of the following function.

$$f(\mathbf{U}; \gamma_r, \gamma_c) \quad = \quad \frac{1}{2}\|\mathcal{P}_\Theta(\mathbf{X}) - \mathcal{P}_\Theta(\mathbf{U})\|_F^2 + \gamma_r J_r(\mathbf{U}) + \gamma_c J_c(\mathbf{U}). \tag{1}$$

The quadratic term quantifies how well $\mathbf{U}$ approximates $\mathbf{X}$ on the observed entries, while the two roughness penalties, $J_r(\mathbf{U})$ and $J_c(\mathbf{U})$, incentivize smoothness across the rows and columns of $\mathbf{U}$. The nonnegative parameters $\gamma_r$ and $\gamma_c$ tune the tradeoff between how well $\mathbf{U}$ agrees with $\mathbf{X}$ over $\Theta$ and how smooth $\mathbf{U}$ is with respect to its rows and columns. By tuning $\gamma_r$ and $\gamma_c$, we obtain estimates of $\mathbf{X}$ at varying scales of row and column smoothness.

In this paper, we use roughness penalties of the following forms

$$J_r(\mathbf{U}) \quad = \quad \sum_{(i,j)\in\mathcal{E}_r} \Omega(\|\mathbf{U}_{i\cdot} - \mathbf{U}_{j\cdot}\|_2) \quad \text{and} \quad J_c(\mathbf{U}) = \sum_{(i,j)\in\mathcal{E}_c} \Omega(\|\mathbf{U}_{\cdot i} - \mathbf{U}_{\cdot j}\|_2), \tag{2}$$

where $\mathbf{U}_{i\cdot}$ ($\mathbf{U}_{\cdot i}$) denotes the $i$th row (column) of the matrix $\mathbf{U}$. The index sets $\mathcal{E}_r$ and $\mathcal{E}_c$ denote the edge sets of row and column graphs that encode a preliminary data-driven assessment of the similarities between rows and columns of the data matrix. The function $\Omega$, which maps $[0, \infty)$ into $[0, \infty)$, will be explained shortly. Variations on the optimization problem of minimizing (1) have been previously proposed in the literature. When there is no data missing, i.e. $\Theta = [m] \times [n]$ and $\Omega$ is a linear mapping, minimizing the objective in (1) produces a convex biclustering problem (Chi et al., 2017). Additionally, if either $\gamma_r$ or $\gamma_c$ is zero, then we obtain convex clustering (Pelckmans et al., 2005; Hocking et al., 2011; Lindsten et al., 2011; Chi & Lange, 2015). If we take $\Omega$ to be a nonlinear concave function, problem (1) reduces to an instance of concave penalized regression-based clustering (Pan et al., 2013; Marchetti & Zhou, 2014; Wu et al., 2016). The convergence properties of our co-clustering procedure will rely on the following two assumptions.

**Assumption 2.1** *The row and column graphs $\mathcal{E}_r$ and $\mathcal{E}_c$ are connected, i.e. the row graph is connected if for any pair of rows, indexed by $i$ and $j$ with $i \neq j$, there exists a sequence of indices $i \to k \to \cdots \to l \to j$ such that $(i, k), \ldots, (l, j) \in \mathcal{E}_r$. A column graph is connected under analogous conditions.*

**Assumption 2.2** *The function $\Omega : [0, \infty) \mapsto [0, \infty)$ is (i) concave and continuously differentiable on $(0, \infty)$, (ii) vanishes at the origin, i.e. $\Omega(0) = 0$, (iii) is increasing on $[0, \infty)$, and (iv) has finite directional derivative at the origin.*

Figure 4 in Appendix A plots an example of a function $\Omega$ that satisfies Assumption 2.2. For concreteness, in the rest of this paper, we use the following function $\Omega$

$$\Omega(z) \quad = \quad \frac{1}{2} \int_0^z \frac{1}{\sqrt{\zeta} + \epsilon} d\zeta, \tag{3}$$

where $\epsilon$ is a small positive number, e.g. $10^{-12}$. The key feature of $\Omega$ in (3), which satisfies Assumption 2.2, is that when it is used in the roughness penalties (2) small differences between rows and columns are penalized significantly more than larger differences resulting in more aggressive smoothing of small noisy variations and leaving intact more significant systematic variations. Specifically, functions that satisfy Assumption 2.2 are not differentiable at the origin and therefore incentivize sparsity in the differences in the rows and columns. Consequently, small noisy variations between pairs of rows and columns are eliminated completely for sufficiently large $\gamma_r$ and $\gamma_c$. This is in contrast with commonly used quadratic penalties. For example, replacing $J_r(\mathbf{U})$ and $J_c(\mathbf{U})$ by quadratic row and column penalties

$$J_r(\mathbf{U}) \quad = \quad \frac{1}{2} \sum_{(i,j)\in\mathcal{E}_r} w_{ij}\|\mathbf{U}_{i\cdot} - \mathbf{U}_{j\cdot}\|_2^2 \quad \text{and} \quad J_c(\mathbf{U}) = \frac{1}{2} \sum_{(i,j)\in\mathcal{E}_c} \tilde{w}_{ij}\|\mathbf{U}_{\cdot i} - \mathbf{U}_{\cdot j}\|_2^2, \tag{4}$$

gives a version of matrix completion on graphs (Kalofolias et al., 2014; Rao et al., 2015), where $w_{ij}$ and $\tilde{w}_{ij}$ are fixed row and column weights inferred from the data. Shahid et al. (2016) also use quadratic row and column penalties to perform joint linear dimension reduction on the rows and columns of a data matrix. Penalties like the ones given in (4), unlike the ones considered in Assumption 2.2, smooth out more significant systematic variations more aggressively and shrink, but do not completely eliminate, small noisy variations.

A simple concrete example illuminates the differences between roughness penalties considered in this paper and commonly used existing penalties. Let $\mathbf{U}_{1.} = (1 \quad 0)^{\mathsf{T}}$ and $\mathbf{U}_{2.} = (1 + \delta \quad 0)^{\mathsf{T}}$ and take $\Omega$ to be as defined in (3). Then,

$$\|\mathbf{U}_{1.} - \mathbf{U}_{2.}\|_2 \quad = \quad \delta, \quad \Omega(\|\mathbf{U}_{1.} - \mathbf{U}_{2.}\|_2) \quad \approx \quad \sqrt{\delta}, \quad \text{and} \quad \|\mathbf{U}_{1.} - \mathbf{U}_{2.}\|_2^2 \quad = \quad \delta^2.$$

Suppose there is a small difference between the first and second rows of $\mathbf{U}$, e.g. $\delta = 10^{-4}$. Then

$$\|\mathbf{U}_{1.} - \mathbf{U}_{2.}\|_2 \quad = \quad 10^{-4}, \quad \Omega(\|\mathbf{U}_{1.} - \mathbf{U}_{2.}\|_2) \quad \approx \quad 10^{-2}, \quad \text{and} \quad \|\mathbf{U}_{1.} - \mathbf{U}_{2.}\|_2^2 \quad = \quad 10^{-8}.$$

Thus, *small* differences are penalized the *most* using the concave $\Omega$ and the *least* by the convex quadratic penalty. Suppose there is a large difference between the first and second rows of $\mathbf{U}$, e.g. $\delta = 10^4$. Then

$$\|\mathbf{U}_{1.} - \mathbf{U}_{2.}\|_2 \quad = \quad 10^4, \quad \Omega(\|\mathbf{U}_{1.} - \mathbf{U}_{2.}\|_2) \quad \approx \quad 10^2, \quad \text{and} \quad \|\mathbf{U}_{1.} - \mathbf{U}_{2.}\|_2^2 \quad = \quad 10^8.$$

Thus, *large* differences are penalized the *least* using the concave $\Omega$ and the *most* by the convex quadratic penalty.

The practical consequence of the differences highlighted by the example above is that the convex penalties, either when $\Omega$ is linear or quadratic, do not introduce enough smoothing for small differences and too much smoothing for large differences. Indeed, (Pan et al., 2013; Marchetti & Zhou, 2014; Wu et al., 2016) showed that superior clustering results could be had, when only $J_r$ or $J_c$ is used, by using concave $\Omega$. Chi et al. (2017) also showed that empirically that the solution to the convex biclustering problem tended to identify too many biclusters and consequently also introduced a one-step reweighted convex biclustering refinement, which recovered the true biclusters more accurately in simulation experiments. The reweighting refinement can be seen as taking a single step in an iterative algorithm for minimizing (1) inexactly when $\Omega$ is concave (Chi et al., 2018; Chi & Steinerberger, 2018). As our method combines biclustering of incomplete data at different scales, or values of $\gamma_r$ and $\gamma_c$, consequently we extend the reweighting refinement introduced by Chi et al. (2017) in Section 2.1.

Our problem formulation (1) is distinct from related problem formulations in the following ways:

1. Rows and columns of $\mathbf{U}$ are *simultaneously* shrunk towards each other as the parameters $\gamma_r$ and $\gamma_c$ increase. Note that this shrinkage procedure is fundamentally different from methods like the clustered dendrogram, which independently cluster the rows and columns as well as alternating partition tree construction procedures (Gavish & Coifman, 2012; Mishne et al., 2016).

2. Our ultimate goal is not to perform matrix completion (though this is a by-product of our approach) but rather to perform joint row and column dimension reduction.

3. Our work generalizes both Shahid et al. (2016) and Chi et al. (2017) in that we seek the flexibility of performing non-linear dimension reduction on the rows and columns of the data matrix, i.e. a more general manifold organization than a co-clustered structure.

4. Instead of determining an optimal single scale of the solution as in Shahid et al. (2016); Chi et al. (2017), we recognize that the multiple scales of the different solutions can be aggregated to better estimate the underlying geometry, similar to the tree-based Earth mover's distance proposed in Ankenman (2014); Mishne et al. (2017).

## 2.1 Co-Clustering Algorithm

We now introduce a majorization-minimization (MM) algorithm (Sun et al., 2017) for solving the minimization in (1). The basic strategy behind an MM algorithm is to convert a hard optimization problem into a sequence of simpler ones. The MM principle requires majorizing the objective

---

**Algorithm 1** CO-CLUSTER-MISSING$(\mathcal{P}_\Theta(\mathbf{X}), \gamma_r, \gamma_c)$

---

1: Initialize $\mathbf{U}_0, \tilde{w}_{r,ij}$, and $\tilde{w}_{c,ij}$
2: **repeat**
3:     $\tilde{\mathbf{X}} \leftarrow \mathcal{P}_\Theta(\mathbf{X}) + \mathcal{P}_{\Theta^c}(\mathbf{U}_t)$
4:     $\{\mathbf{U}_{t+1}, n_r, n_c\} \leftarrow$ CONVEX-BICLUSTER $\left( \tilde{\mathbf{X}}, \gamma_r, \gamma_c, \{\tilde{w}_{r,ij}\}, \{\tilde{w}_{c,ij}\} \right)$
5:     $\tilde{w}_{r,ij} \leftarrow \Omega'(\|\mathbf{U}_{t+1,i\cdot} - \mathbf{U}_{t+1,j\cdot}\|_2)$ for all $(i,j) \in \mathcal{E}_r$
6:     $\tilde{w}_{c,ij} \leftarrow \Omega'(\|\mathbf{U}_{t+1,\cdot i} - \mathbf{U}_{t+1,\cdot j}\|_2)$ for all $(i,j) \in \mathcal{E}_c$
7: **until** convergence
8: Return $\left\{ \mathbf{U}(\gamma_r, \gamma_c) = \mathbf{U}_t, \tilde{\mathbf{X}}, n_r, n_c \right\}$

---

function $f(\mathbf{U})$ by a surrogate function $g(\mathbf{U} \mid \tilde{\mathbf{U}})$ anchored at $\tilde{\mathbf{U}}$. Majorization is a combination of the tangency condition $g(\mathbf{U} \mid \tilde{\mathbf{U}}) = f(\tilde{\mathbf{U}})$ and the domination condition $g(\mathbf{U} \mid \tilde{\mathbf{U}}) \geq f(\mathbf{U})$ for all $\mathbf{U} \in \mathbb{R}^{m \times n}$. The associated MM algorithm is defined by the iterates $\mathbf{U}_{t+1} = \arg\min_{\mathbf{U}} g(\mathbf{U} \mid \mathbf{U}_t)$.

It is straightforward to verify that the MM iterates generate a descent algorithm driving the objective function downhill, i.e. that $f(\mathbf{U}_{t+1}) \leq f(\mathbf{U}_t)$ for all $t$.

The following function

$$g(\mathbf{U} \mid \tilde{\mathbf{U}}) = \frac{1}{2} \|\tilde{\mathbf{X}} - \mathbf{U}\|_F^2 + \gamma_r \sum_{(i,j) \in \mathcal{E}_r} \tilde{w}_{r,ij} \|\mathbf{U}_{i\cdot} - \mathbf{U}_{j\cdot}\|_2 + \gamma_c \sum_{(i,j) \in \mathcal{E}_c} \tilde{w}_{c,ij} \|\mathbf{U}_{\cdot i} - \mathbf{U}_{\cdot j}\|_2 + \kappa$$

majorizes our objective function (1) at $\tilde{\mathbf{U}}$, where $\kappa$ is a constant that does not depend on $\mathbf{U}$ and $\tilde{w}_{r,ij}$ and $\tilde{w}_{c,ij}$ are weights that depend on $\tilde{\mathbf{U}}$, i.e.

$$\tilde{w}_{r,ij} = \Omega'(\|\tilde{\mathbf{U}}_{i\cdot} - \tilde{\mathbf{U}}_{j\cdot}\|_2) \quad \text{and} \quad \tilde{w}_{c,ij} = \Omega'(\|\tilde{\mathbf{U}}_{\cdot i} - \tilde{\mathbf{U}}_{\cdot j}\|_2), \tag{5}$$

where $\Omega'$ denotes the first derivative of $\Omega$. We give a detailed derivation of this majorization in Appendix A.

Minimizing $g(\mathbf{U} \mid \tilde{\mathbf{U}})$ is equivalent to minimizing the objective function of the convex biclustering problem for which efficient algorithms have been introduced (Chi et al., 2017). Thus, in the $t + 1$th iteration, our MM algorithm solves a convex biclustering problem where the missing values in $\mathbf{X}$ have been replaced with the values of $\tilde{\mathbf{U}} = \mathbf{U}_t$ and the weights $\tilde{w}_{r,ij}$ and $\tilde{w}_{c,ij}$ have been computed based on $\tilde{\mathbf{U}} = \mathbf{U}_t$ according to (5). Note that the weights are continuously updated throughout the optimization as opposed to the fixed weights in (Chi et al., 2017). This introduces a notion of the scale of the solution into the weights.

Algorithm 1 summarizes our MM algorithm, CO-CLUSTER-MISSING, which returns a smooth output matrix $\mathbf{U}(\gamma_r, \gamma_c)$, a filled-in matrix $\tilde{\mathbf{X}} = \mathcal{P}_\Theta(\mathbf{X}) + \mathcal{P}_{\Theta^c}(\mathbf{U}(\gamma_r, \gamma_c))$ as well as $n_r$ and $n_c$, which are respectively the number of distinct rows and distinct columns in $\mathbf{U}(\gamma_r, \gamma_c)$. The CO-CLUSTER-MISSING algorithm has the following convergence guarantee.

**Proposition 1** *Under Assumption 2.1 and Assumption 2.2, the sequence $\mathbf{U}_t$ generated by Algorithm 1 has at least one limit point, and all limit points are stationary points of (1).*

The proof of Proposition 1 is presented in Appendix B.

## 2.2 CO-CLUSTERING AT MULTIPLE SCALES

Initializing Algorithm 1 is very important as the objective function in (1) is not convex. The matrix $\mathbf{U}^{(0)}$ is initialized to be the mean of all non-missing values. The connectivity graphs $\mathcal{E}_r$ and $\mathcal{E}_c$ are initialized at the beginning using $k$-nearest-neighbor graphs, and remain fixed throughout all considered scales. If we observed the complete matrix, employing a sparse Gaussian kernel is a natural way to quantify the local similarity between pairs of rows and pairs of columns. The challenge is that we do not have the complete data matrix $\mathbf{X}$ but only the partially observed one $\mathcal{P}_\Theta(\mathbf{X})$. Therefore, we rely only on the observed values to calculate the $k$-nearest-neighbor graph, based on the distance used by Ram et al. (2013) in an image inpainting problem.

---

**Algorithm 2** Co-manifold learning on an Incomplete Data Matrix

---

1: Initialize $\mathcal{E}_r, \mathcal{E}_c$
2: Set $d(\mathbf{X}_{\cdot i}, \mathbf{X}_{\cdot j}) = 0$ and $d(\mathbf{X}_{\cdot i}, \mathbf{X}_{\cdot j}) = 0$
3: Set $n_r = m, n_c = n, k = k_0$, and $l = l_0$
4: **while** $n_r > 1$ **do**
5:    **while** $n_c > 1$ **do**
6:      $\left\{ \mathbf{U}^{(l,k)}, \tilde{\mathbf{X}}^{(l,k)}, n_r, n_c \right\} \leftarrow \text{CO-CLUSTER-MISSING}\left( \mathcal{P}_\Theta(\mathbf{X}), \gamma_r = 2^l, \gamma_c = 2^k \right)$
7:      Update row distances: $d\left( \mathbf{X}_{i\cdot}, \mathbf{X}_{j\cdot} \right) \mathrel{+}= d\left( \tilde{\mathbf{X}}_{i\cdot}^{(l,k)}, \tilde{\mathbf{X}}_{j\cdot}^{(l,k)} \right)$
8:      Update column distances: $d\left( \mathbf{X}_{\cdot i}, \mathbf{X}_{\cdot j} \right) \mathrel{+}= d\left( \tilde{\mathbf{X}}_{\cdot i}^{(l,k)}, \tilde{\mathbf{X}}_{\cdot j}^{(l,k)} \right)$
9:      $k \leftarrow k + 1$
10:    **end while**
11:    $l \leftarrow l + 1$
12: **end while**
13: Calculate affinities $\mathbf{A}_r(\mathbf{X}_{i\cdot}, \mathbf{X}_{j\cdot})$ and $\mathbf{A}_c(\mathbf{X}_{\cdot i}, \mathbf{X}_{\cdot j})$
14: Calculate embeddings $\Psi_r, \Psi_c$

---

Solving the co-clustering problem in Algorithm 1 at a single scale yields the smooth estimate $\mathbf{U}$, the filled-in data matrix $\tilde{\mathbf{X}}$, and $n_r$ and $n_c$ which are the number of distinct row and column clusters, respectively, identified at that scale through columns and rows merging in $\mathbf{U}$. To obtain a collection of estimates at multiple scales, we solve the optimization problem for pairs of values for $\gamma_r, \gamma_c$ set at logarithmic scale (Chi & Steinerberger, 2018) until we have converged to single global bi-cluster, i.e. $n_r = n_c = 1$.

We start with small values of $\gamma_r = 2^{l_0}$ and $\gamma_c = 2^{k_0}$, where $l_0, k_0 < 0$. We apply the co-clustering (Algorithm 1) and obtain the smooth estimate $\mathbf{U}^{(l_0,k_0)} = \mathbf{U}(2^{l_0}, 2^{k_0})$ used to fill in the data matrix $\tilde{\mathbf{X}}^{(l_0,k_0)}$. Keeping $\gamma_r$ fixed, we continue increasing $\gamma_c$ by power of 2 and applying the biclustering until the algorithm converges to one cluster along the columns ($n_c = 1$). We then increase $\gamma_r$ by power of 2 and reset $\gamma_c = 2^{k_0}$. We repeat this procedure at increasing scales of $\gamma_r = 2^l, \quad \gamma_c = 2^k$, until we have converged to a single global bicluster. This multiscale procedure yields a collection of filled-in matrices at all scales $\left\{ \tilde{\mathbf{X}}^{(l,k)} \right\}_{l,k}$. Note that the $l$ and $k$ denote the power of 2 taken for specific row and column cost parameters ($\gamma_c, \gamma_r$) in the solution. This is intended as a compact notation that corresponds a pair of parameters ($\gamma_r, \gamma_c$) to their solution $\mathbf{U}^{(l,k)}$ and filled in estimate $\tilde{\mathbf{X}}^{(l,k)}$.

## 3 CO-MANIFOLD LEARNING

Kernel-based manifold learning relies on constructing a good similarity measure between points, and a dimension reduction method based on this similarity. The eigenvectors of these kernels are typically used as the new low-dimensional coordinates for the data. Here we leverage having calculated an estimate of the filled-in matrix at multiple scales $\left\{ \tilde{\mathbf{X}}^{(l,k)} \right\}_{l,k}$, to define a new metric between rows and columns. This metric will encompass all bi-scales as defined by joint pairs of optimization cost parameters $\gamma_r, \gamma_c$. Given a new metric we employ Diffusion maps (Coifman & Lafon, 2006) to obtain a new embedding of the rows and columns. The full algorithm is given in Algorithm 2.

### 3.1 MULTI-SCALE METRIC

We define a new metric to estimate the geometry of the complete data matrix both locally and globally. For a given pair $\gamma_r, \gamma_c$, we calculate the Euclidean distance between rows for the filled-in matrix at that joint scale, weighted by the cost parameters:

$$d\left( \tilde{\mathbf{X}}_{i\cdot}^{(l,k)}, \tilde{\mathbf{X}}_{j\cdot}^{(l,k)} \right) \quad = \quad (\gamma_r \gamma_c)^\alpha \| \tilde{\mathbf{X}}_{i\cdot}^{(l,k)} - \tilde{\mathbf{X}}_{j\cdot}^{(l,k)} \|_2$$

where $\tilde{\mathbf{X}}^{(l,k)} = \mathcal{P}_\Theta(\mathbf{X}) + \mathcal{P}_{\Theta^c}(\mathbf{U}^{(l,k)})$, and we set $\alpha = -1/2$ to favor local over global structure in our simulations. Our goal is to aggregate distances between a pair of rows (columns) across multiple scales of the solution, to calculate a metric that better recovers the local and global geometry of the data despite the missing values, thus "fixing" the missing data metric. Having solved for multiple pairs from the solution surface, we sum over all the distances to obtain a multi-scale distance on the data rows:

$$d(\mathbf{X}_{i\cdot}, \mathbf{X}_{j\cdot}) \quad = \quad \sum_{l,k} d\left( \tilde{\mathbf{X}}_{i\cdot}^{(l,k)}, \tilde{\mathbf{X}}_{j\cdot}^{(l,k)} \right).$$

An analogous multi-scale distance is computed for pairs of columns. Note that if there are no missing values, this metric is just the Euclidean pairwise distance scaled by a scalar, so that we recover the original embedding of the complete matrix.

This metric takes advantage of solving the optimization for multiple pairs of cost parameters and filling in the missing values with increasingly smooth estimates (as $\gamma_r$ and $\gamma_c$ increase). It also alleviates the need to identify the ideal scale at which to fill in the points; it is not clear that a single "optimal" scale actually exists, but rather different points in the matrix may have different optimal scales. As opposed to the partition-tree based metric of Mishne et al. (2017), this metric takes into account all joint scales of the data as the matrix $\mathbf{U}$ is smoothed across rows and columns simultaneously, thus fully taking advantage of the coupling between both modes.

## 3.2 DIFFUSION MAPS

Having calculated a multi-scale metric on the rows and columns throughout the joint optimization procedure, we can now construct a pair of low-dimensional embeddings based on these distances. Specifically we use Diffusion maps (Coifman & Lafon, 2006), but any dimension reduction technique relying on the construction of a distance kernel could be used instead. We briefly review the construction of the diffusion maps for the rows (features) of a matrix but the same can be applied to the columns (observations). Given a distance between two rows of the matrix $d(\mathbf{X}_{i\cdot}, \mathbf{X}_{j\cdot})$, we construct an affinity kernel on the rows. We choose an exponential function, but other kernels can be considered depending on the application:

$$\mathbf{A}[i,j] \quad = \quad \exp\{-d^2(\mathbf{X}_{i\cdot}, \mathbf{X}_{j\cdot})/\sigma^2\},$$

where $\sigma$ is a scale parameter. The exponential function enhances locality, as pairs of samples whose distance exceed $\sigma$ have negligible affinity. One possible choice for $\sigma$ is to be the median of distances within the data.

We derive a row-stochastic matrix $\mathbf{P}$ by normalizing the rows of matrix $\mathbf{A}$: $\mathbf{P} = \mathbf{D}^{-1}\mathbf{A}$, where $\mathbf{D}$ is a diagonal matrix whose elements are given by $\mathbf{D}[i,i] = \sum_j \mathbf{A}[i,j]$. The eigendecomposition of $\mathbf{P}$ yields a sequence of positive decreasing eigenvalues: $1 = \lambda_0 \geq \lambda_1 \geq \ldots$, and right eigenvectors $\{\psi_\ell\}_\ell$. Retaining only the first $d$ eigenvalues and eigenvectors, the mapping $\Psi$ embeds the rows into the Euclidean space $\mathbb{R}^d$:

$$\Psi : \mathbf{X}_{i\cdot} \rightarrow \left( \lambda_1 \psi_1(i), \lambda_2 \psi_2(i), \ldots, \lambda_d \psi_d(i) \right)^\mathsf{T}.$$

The embedding integrates the local connections found in the data into a global representation, which enables visualization of the data, organizes the data into meaningful clusters, and identifies outliers and singular samples. This embedding is also equipped with a noise-robust distance, the diffusion distance. For more details on diffusion maps, see Coifman & Lafon (2006).

## 4 NUMERICAL EXPERIMENTS

The model we consider in the paper is such that the data is not represented by a biclustering model but rather at least one of the modes (rows/columns) lies on a low-dimensional manifold. In our experiments we consider three such examples. In the first a manifold structure exists along both rows and columns, and for the second and third the columns belong to disjoint clusters while the rows lie on a manifold:

- **linkage** A synthetic dataset with a one-dimensional manifold along the rows and a two-dimensional manifold along the columns. Let $\{z_i\}_{i=1}^{N_1} \in \mathbb{R}^3$ be points along a helix and

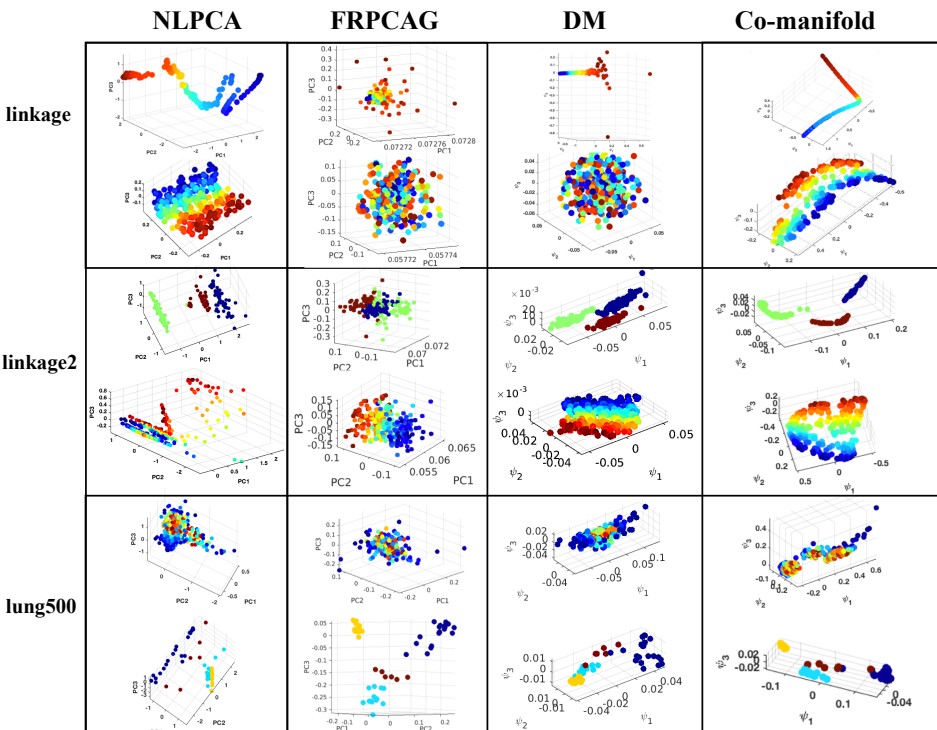

Figure 2: Comparing row and column embeddings of NLPCA, FRPCAG, DM, Ours, for three datasets with 50% missing entries. For each dataset, top / bottom row is embedding of rows / columns of $\mathbf{X}$. For the **lung500** dataset, the color of the clusters are as follows: yellow - normal subjects, dark blue - carcinoid, cyan - colon , red - small cell carcinoma.

let $\{y_j\}_{j=1}^{N_2} \in \mathbb{R}^3$ be a two dimensional surface, where we set $N_1 = 190$, $N_2 = 300$. We analyze the matrix of Euclidean distances between the two spatially distant sets of points to reveal the underlying geometry of both rows and columns,

$$\mathbf{X}[i,j] = \|z_i - y_j\|_2. \tag{6}$$

Other functions of the distance can also be used such as the elastic or Coulomb potential operator (Coifman & Gavish, 2011). Missing values correspond to having access to only some of the distances between pairs of points across the two sets. Note that this is unlike MDS as we do not have pairwise distances between all datapoints, but rather distances between two sets of points with different geometries.

- **linkage2** A synthetic dataset with a clustered structure along the rows and a two-dimensional manifold along the columns. Let $\{x_i\}_{i=1}^{N_1} \in \mathbb{R}^3$ be composed of points in 3 Gaussian clouds in 3D and let $\{y_j\}_{j=1}^{N_2} \in \mathbb{R}^3$ be a two dimensional surface as before, where we set $N_1 = 200$, $N_2 = 300$.

- **lung500** A real-world dataset composed of 56 lung cancer patients and their gene expression (Lee et al., 2010). We selected the 500 genes with the greatest variance from the original collection of 12,625 genes. Subjects belong to one of four subgroups; they are either normal subjects (Normal) or have been diagnosed with one of three types of cancers: pulmonary carcinoid tumors (Carcinoid), colon metastases (Colon), and small cell carcinoma (Small Cell).

For all datasets, the rows and columns of the data matrix are randomly permuted so their natural order does not play a role in inferring the geometry. We evaluate results both qualitatively and quantitatively.

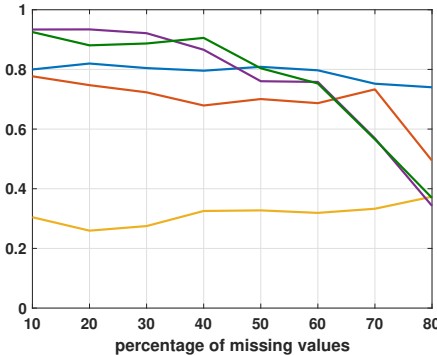 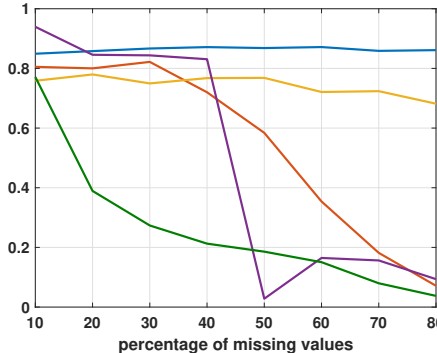

Figure 3: Comparing $k$-means clustering applied to embedding of data for increasing percentages of missing values for our multiscale approach (blue), diffusion maps of missing data matrix (red), NLPCA (yellow) and FRPCAG for two different cost parameter settings (green and purple). We evaluate using the adjusted Rand Index (ARI) compared to the ground-truth labels of (left) the 4 cancer types for the **lung500** dataset, and (right) 3 Gaussian clusters of the **linkage2** dataset.

We compare our embeddings to three approaches: NLPCA with missing data completion (Scholz et al., 2005), Fast Robust PCA on Graphs (FRPCAG) (Shahid et al., 2016) and Diffusion maps (DM) (Coifman & Lafon, 2006) on the missing data. FRPCAG provides a *linear* embedding of a low-rank estimate of a data matrix using quadratic row and column penalties, while the other methods output nonlinear embeddings. NLPCA and DM are applied to each mode separately, while our method and FRPCAG take into account the coupled geometry. Comparing to Diffusion maps demonstrates how missing values corrupt the embedding. Note that this is also equivalent to applying our approach for only a single scale of the cost parameters ($\gamma_r, \gamma_c \to \infty$), as for this choice of parameters the solution $\mathbf{U}$ converges to the grand mean of the data.

In Figure 2, we display the embeddings of the different methods for each of the three datasets for both their rows (top) and their columns (bottom), where 50% of the entries have been removed. Both NLPCA and DM reveal the underlying 2D surface structure on the rows in only one of the linkage datasets, and err greatly on the other. DM correctly infers a 1D path for the **linkage** dataset but it is increasingly noisy. For NLPCA the 1D embedding is not as smooth and clean as the embedding inferred by the co-manifold approach. Our method reveals the 2D surface in both cases. FRPCAG is unsuccessful in uncovering the non-linear manifolds underlying either rows or columns in the **linkage** dataset. For **linkage2**, the rows do not separate cleanly into disjoint clusters, and for the columns only one of the parameters of the 2D plane is uncovered.

For the **lung500** data, NLPCA and DM embed the cancer samples such that the normal subjects (yellow) are close to the Colon type (cyan), whereas both our method and FRPCAG separates the normal subjects from the cancer types. This is due to taking into account the coupled structure of the genes and the samples. As opposed to the clustered structure along the samples, the three non-linear methods reveal a smooth manifold structure to the genes, which is different than the assumed clustered structure a biclustering method would infer. The representation yielded by FRPCAG is the least structured but also does not reveal disjoint gene clusters. For plots presenting the datasets and filled-in values at multiple scales see Appendix C.

Manifold learning is not used only for data visualization but also for calculating new data representations that can then be used for signal processing and machine learning tasks. Here we calculate clustering accuracy of clustering the low-dimensional representation of the data. We apply $k$-means to the column embeddings of each method, with $k$ set to the correct number of clusters in the data, as we want to evaluate the ability of the methods to properly represent the data without being sensitive to empirical estimation of the number of clusters in the data. We use the Adjusted Rand Index (ARI) (Hubert & Arabie, 1985), to measure the similarity between the $k$-means clustering of the embedding and the ground-truth labels.

We note that Shahid et al. (2016) do not provide guidelines by which to select the appropriate cost parameters $\gamma_r$ and $\gamma_c$ in the FRPCAG solution. Therefore we calculated the solution using a wide range of possible values and plot the results for worst and best performance to demonstrate the

sensitivity to setting these parameters. In contrast, our approach alleviates the need to perform this parameter selection.

The left panel of Figure 3 compares clustering the embedding of the cancer patients in **lung500** by each method for increasing percentage of missing values in the data, where we averaged over 30 realizations of missing entries. For low values of missing data, FRPCAG performs the best but its performance degrades as the percentage of missing values increases. For higher values of missing data (50% and above), our embedding (blue plot) gives the best clustering result and its performance is only slightly degraded by increasing the percentage of missing values, as opposed to Diffusion maps (red plot). This demonstrates that the metric we calculate is a good estimate of the metric of the complete data matrix. NLPCA (yellow plot) performs worst.

The right panel of Figure 3 compares clustering the embedding of the three Gaussian clusters in **linkage2** for increasing percentage of missing values in the data, where we averaged over 30 realizations of the data itself and the missing entries. Our embedding (blue plot) gives the best clustering result (for 20% missing values and above) and its performance is unaffected by increasing the percentage of missing values up to 80%, as opposed to Diffusion maps (red plot) which is greatly degraded by the missing values. NLPCA (yellow plot) does not perform as well as our approach, with performance decreasing as the percentage of missing values increases. For a good parameter selection FRPCAG (purple plot) performs well for low percentage of missing values, and then performance is drastically impaired above $50\%$. For a poor parameter selection FRPCAG (green plot) has the worst performance for practically all percentages of missing values. Note that the overall poor performance of FRPCAG in this setting is due to the underlying manifold being non-linear and the data being high-rank, demonstrating the need for nonlinear embedding approaches.

## 5 Conclusions

In this paper we presented a new method for learning nonlinear manifold representations of both the rows and columns of a matrix with missing data. We proposed a new optimization problem to obtain a smooth estimate of the missing data matrix, and solved this problem for different values of the cost parameters, which encode the smoothness scale of the estimate along the rows and columns. We leverage calculating these multi-scale estimates into a new metric that aims to capture the geometry of the complete data matrix. This metric is then used in a kernel-based manifold learning technique to obtain new representations of both the rows and the columns. In future work we will investigate additional metrics in a general co-manifold setting and relate them to optimal transport problem and Earth Mover's Distance (Coifman & Leeb, 2013). We also intend to develop efficient solutions to accelerate the optimization in order to address large-scale datasets, in addition to the small-scale regime we demonstrate here. We note that the datasets considered here, while being small-scale in the observation domain are high-dimensional in the feature domain, which is a non-trivial setting, and indeed a challenge for supervised methods such as deep learning due to limited training data.

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

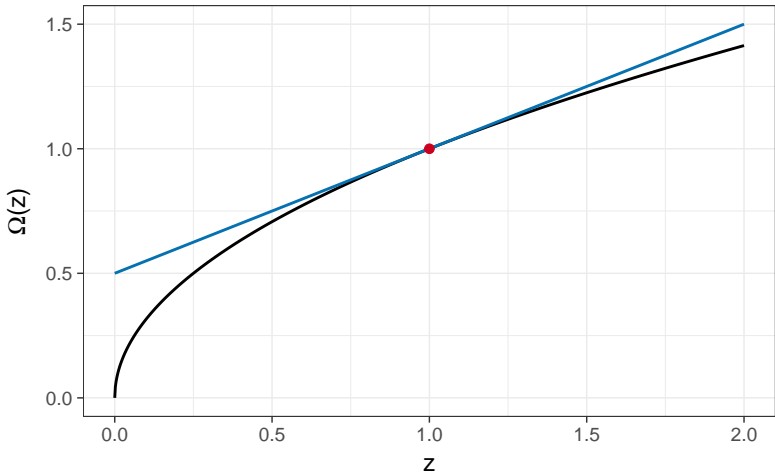

Figure 4: Majorization of the $\Omega$ function (black) given in (3) by its first-order Taylor approximation at 1 (blue).

## A DERIVATION OF MAJORIZATION

We first construct a majorization of the data-fidelity term. It is easy to verify that the following function of $\mathbf{U}$

$$g_1(\mathbf{U} \mid \tilde{\mathbf{U}}) = \frac{1}{2} \|\tilde{\mathbf{X}} - \mathbf{U}\|_{\mathrm{F}}^2, \tag{7}$$

where $\tilde{\mathbf{X}} = \mathcal{P}_\Theta(\mathbf{X}) + \mathcal{P}_{\Theta^c}(\tilde{\mathbf{U}})$, majorizes the data-fidelity term $\frac{1}{2}\|\mathcal{P}_\Theta(\mathbf{X}) - \mathcal{P}_\Theta(\mathbf{U})\|_{\mathrm{F}}^2$ at $\tilde{\mathbf{U}}$.

We next construct a majorization of the penalty term. Recall that the first-order Taylor approximation of a differentiable concave function provides a global upper bound on the function. Therefore, under Assumption 2.2, we have the following inequality

$$\Omega(z) \leq \Omega(\tilde{z}) + \Omega'(\tilde{z})(z - \tilde{z}), \qquad \text{for all } z, \tilde{z} \in [0, \infty).$$

Figure 4 shows the relationship between $\Omega$ given in (3) with $\epsilon = 10^{-12}$ and its first-order Taylor approximation at $\tilde{z} = 1$.

Thus, we can majorize the penalty term $\gamma_r J_r(\mathbf{U}) + \gamma_c J_c(\mathbf{U})$ with the function

$$g_2(\mathbf{U} \mid \tilde{\mathbf{U}}) = \gamma_r \sum_{(i,j) \in \mathcal{E}_r} \tilde{w}_{r,ij} \|\mathbf{U}_{i\cdot} - \mathbf{U}_{j\cdot}\|_2 + \gamma_c \sum_{(i,j) \in \mathcal{E}_c} \tilde{w}_{c,ij} \|\mathbf{U}_{\cdot i} - \mathbf{U}_{\cdot j}\|_2 + \kappa, \tag{8}$$

where $\kappa$ is a constant that does not depend on $\mathbf{U}$ and $\tilde{w}_{r,ij}$ and $\tilde{w}_{c,ij}$ (5) are weights that depend on $\tilde{\mathbf{U}}$. The sum of functions (7) and (8)

$$
\begin{aligned}
g(\mathbf{U} \mid \tilde{\mathbf{U}}) &= g_1(\mathbf{U} \mid \tilde{\mathbf{U}}) + g_2(\mathbf{U} \mid \tilde{\mathbf{U}}) \\
&= \frac{1}{2} \|\tilde{\mathbf{X}} - \mathbf{U}\|_{\mathrm{F}}^2 + \gamma_r \sum_{(i,j) \in \mathcal{E}_r} \tilde{w}_{r,ij} \|\mathbf{U}_{i\cdot} - \mathbf{U}_{j\cdot}\|_2 + \gamma_c \sum_{(i,j) \in \mathcal{E}_c} \tilde{w}_{c,ij} \|\mathbf{U}_{\cdot i} - \mathbf{U}_{\cdot j}\|_2 + \kappa
\end{aligned}
\tag{9}
$$

majorizes our objective function (1) at $\tilde{\mathbf{U}}$.

## B PROOF OF PROPOSITION 1

The MM algorithm generates a sequence of iterates that has at least one limit point, and the limit points are stationary points of the objective function

$$f(\mathbf{U}) = \frac{1}{2} \|\mathcal{P}_\Theta(\mathbf{X}) - \mathcal{P}_\Theta(\mathbf{U})\|_{\mathrm{F}}^2 + \gamma_r J_r(\mathbf{U}) + \gamma_c J_c(\mathbf{U}). \tag{10}$$

To reduce notational clutter, we suppress the dependency of $f$ on $\gamma_r$ and $\gamma_c$ since they are fixed during Algorithm 1. We prove Proposition 1 in three stages. First, we show that all limit points of the MM algorithm are fixed points of the MM algorithm map. Second, we show that fixed points of the MM algorithm are stationary points of $f$ in (10). Finally, we show that the MM algorithm has at least one limit point.

## B.1    LIMIT POINTS ARE FIXED POINTS

The convergence theory of MM algorithms relies on the properties of the algorithm map $\psi(\mathbf{U})$ that returns the next iterate given the last iterate. For easy reference, we state a simple version of Meyer's monotone convergence theorem Meyer (1976), which is instrumental in proving convergence in our setting.

**Theorem 1** *Let $f(\mathbf{U})$ be a continuous function on a domain $S$ and $\psi(\mathbf{U})$ be a continuous algorithm map from $S$ into $S$ satisfying $f(\psi(\mathbf{U})) < f(\mathbf{U})$ for all $\mathbf{U} \in S$ with $\psi(\mathbf{U}) \neq \mathbf{U}$. Then all limit points of the iterate sequence $\mathbf{U}_k = \psi(\mathbf{U}_{k-1})$ are fixed points of $\psi(\mathbf{U})$.*

In order to apply Theorem 1, we need to identify elements in the assumption with specific functions and sets corresponding to the problem of minimizing (10). Throughout the following proof, it will sometimes be convenient to work with the column major vectorization of a matrix. The vector $\mathbf{b} = \mathrm{vec}(\mathbf{B})$ is obtained by stacking the columns of $\mathbf{B}$ on top of each other.

**The function $f$:** Take $\mathcal{S} = \mathbb{R}^{m \times n}$ and $f : \mathcal{S} \mapsto \mathbb{R}$ to be the objective function in (10) and majorize $f$ with $g(\mathbf{U} \mid \tilde{\mathbf{U}})$ given in (9). The function $f$ is continuous. Let $\psi(\tilde{\mathbf{U}}) = \arg\min_{\mathbf{U}} g(\mathbf{U} \mid \tilde{\mathbf{U}})$ denote the algorithm map for the MM algorithm. Since $g(\mathbf{U} \mid \tilde{\mathbf{U}})$ is strongly convex in $\mathbf{U}$, it has a unique global minimizer. Consequently, $f(\psi(\mathbf{U})) < f(\mathbf{U})$ for all $\psi(\mathbf{U}) \neq \mathbf{U}$.

**Continuity of the algorithm map $\psi$:** Continuity of $\psi$ follows from the fact that the solution to the convex biclustering problem is jointly continuous in the weights and data matrix (Chi et al., 2017)[Proposition 2]. The weight $\tilde{w}_{r,ij}(\tilde{\mathbf{U}}) = \Omega'(\|\mathbf{U}_{i\cdot} - \mathbf{U}_{j\cdot}\|_2)$ is a continuous function of $\tilde{\mathbf{U}}$, since $\Omega'$ is continuous according to Assumption 2.2. The weight $\tilde{w}_{c,ij}(\tilde{\mathbf{U}})$ is likewise continuous in $\tilde{\mathbf{U}}$. The data matrix passed into the convex biclustering algorithm is $\tilde{\mathbf{X}} = \mathcal{P}_\Theta(\mathbf{X}) + \mathcal{P}_{\Theta^c}(\tilde{\mathbf{U}})$, which is a continuous function of $\tilde{\mathbf{U}}$ since the projection mapping $\mathcal{P}_{\Theta^c}$ is continuous.

## B.2    FIXED POINTS ARE STATIONARY POINTS

Let $\mathbf{L}_{ij} = (\mathbf{e}_i - \mathbf{e}_j)^\mathsf{T} \otimes \mathbf{I}$ and $\tilde{\mathbf{L}}_{ij} = \mathbf{I} \otimes (\mathbf{e}_i - \mathbf{e}_j)^\mathsf{T}$, where $\otimes$ denotes the Kronecker product. Then

$$\mathrm{vec}(\mathbf{U}_{i\cdot} - \mathbf{U}_{j\cdot}) \;\; = \;\; \mathbf{L}_{ij}\mathbf{u} \qquad \text{and} \qquad \mathrm{vec}(\mathbf{U}_{\cdot i} - \mathbf{U}_{\cdot j}) \;\; = \;\; \tilde{\mathbf{L}}_{ij}\mathbf{u}.$$

The directional derivative of $f$ in the direction $\mathbf{v}$ at a point $\mathbf{u}$ is given by

$$\Omega'(\|\mathbf{L}_{ij}\mathbf{u}\|_2; \mathbf{v}) \;\; = \;\; \begin{cases} \Omega'(\|\mathbf{L}_{ij}\mathbf{u}\|_2)\langle \mathbf{L}_{ij}\mathbf{v}, \frac{\mathbf{L}_{ij}\mathbf{u}}{\|\mathbf{L}_{ij}\mathbf{u}\|_2}\rangle & \mathbf{L}_{ij}\mathbf{u} \neq \mathbf{0} \\ \Omega'(\|\mathbf{L}_{ij}\mathbf{u}\|_2)\|\mathbf{L}_{ij}\mathbf{v}\|_2 & \text{otherwise.} \end{cases}$$

A point $\mathbf{u}$ is a stationary point of $f$, if for all direction vectors $\mathbf{v}$

$$0 \;\; \leq \;\; \langle \mathcal{P}_\Theta(\mathbf{u} - \mathbf{x}), \mathbf{v}\rangle + \gamma_r \sum_{(i,j) \in \mathcal{E}_r} \Omega'(\|\mathbf{L}_{ij}\mathbf{u}\|_2; \mathbf{v}) + \gamma_c \sum_{(i,j) \in \mathcal{E}_c} \Omega'(\|\tilde{\mathbf{L}}_{ij}\mathbf{u}\|_2; \mathbf{v}),$$

where $\mathcal{P}_\Theta(\mathbf{u} - \mathbf{x}) = \mathrm{vec}(\mathcal{P}_\Theta(\mathbf{U}) - \mathcal{P}_\Theta(\mathbf{X}))$.

A point $\mathbf{u}$ is a fixed point of $\psi$, if $\mathbf{0}$ is in the subdifferential of $g(\mathbf{u} \mid \mathbf{u})$, i.e.

$$\mathbf{0} \in \{\mathcal{P}_\Theta(\mathbf{u} - \mathbf{x})\} + \gamma_r \sum_{(i,j) \in \mathcal{E}_r} \Omega'(\|\mathbf{L}_{ij}\mathbf{u}\|_2)\partial\|\mathbf{L}_{ij}\mathbf{u}\|_2 + \gamma_c \sum_{(i,j) \in \mathcal{E}_c} \Omega'(\|\tilde{\mathbf{L}}_{ij}\mathbf{u}\|_2)\partial\|\tilde{\mathbf{L}}_{ij}\mathbf{u}\|_2, \quad (11)$$

where the set on the right is the subdifferential $\partial g(\mathbf{u} \mid \mathbf{u})$.

If $\mathbf{L}_{ij}\mathbf{u} \neq \mathbf{0}$, then $\partial\|\mathbf{L}_{ij}\mathbf{u}\|_2 = \left\{\mathbf{L}_{ij}^\mathsf{T} \frac{\mathbf{L}_{ij}\mathbf{u}}{\|\mathbf{L}_{ij}\mathbf{u}\|_2}\right\}$. On the other hand, if $\mathbf{L}_{ij}\mathbf{u} = \mathbf{0}$, then $\partial\|\mathbf{L}_{ij}\mathbf{u}\|_2 = \partial\|\mathbf{0}\|_2 = \{\mathbf{d} : \|\mathbf{d}\|_2 \leq 1\}$.

Fix an arbitrary direction vector $\mathbf{v}$. The inner product of $\mathbf{v}$ with an element in the set on right hand side of (11) is given by

$$\langle \mathcal{P}_\Theta(\mathbf{u} - \mathbf{x}), \mathbf{v} \rangle + \gamma_r \sum_{(i,j) \in \mathcal{E}_r} \Omega'(\|\mathbf{L}_{ij}\mathbf{u}\|_2)\langle \mathbf{d}_{ij}, \mathbf{v} \rangle + \gamma_c \sum_{(i,j) \in \mathcal{E}_c} \Omega'(\|\tilde{\mathbf{L}}_{ij}\mathbf{u}\|_2)\langle \mathbf{d}_{ij}, \mathbf{v} \rangle, \qquad (12)$$

where $\mathbf{d}_{ij} \in \partial\|\mathbf{L}_{ij}\mathbf{u}\|_2$ and $\tilde{\mathbf{d}}_{ij} \in \partial\|\tilde{\mathbf{L}}_{ij}\mathbf{u}\|_2$.

Then the supremum of the right hand side of (12) over all $\mathbf{d}_{ij} \in \partial\|\mathbf{L}_{ij}\mathbf{u}\|_2$ and $\tilde{\mathbf{d}}_{ij} \in \partial\|\tilde{\mathbf{L}}_{ij}\mathbf{u}\|_2$ is nonnegative, because $\mathbf{0} \in \partial g(\mathbf{u} \mid \mathbf{u})$. Consequently, all fixed points of $\psi$ are also stationary points of $f$.

## B.3 THE MM ITERATE SEQUENCE HAS A LIMIT POINT

To ensure the existence of a limit point, we show that the function $f$ is coercive, i.e. $f(\mathbf{U}_t) \to \infty$ for any sequence $\|\mathbf{U}_t\|_F \to \infty$. Recall that according to Assumption 2.1 we assume that the row and column edge sets $\mathcal{E}_r$ and $\mathcal{E}_c$ form connected graphs. Therefore, $J_r(\mathbf{U}) = J_c(\mathbf{U}) = 0$ if and only if $\mathbf{U} = a\mathbf{1}\mathbf{1}^\mathsf{T}$ (Chi et al., 2017, Proposition 3). The edge-incidence matrix of the column graph $\boldsymbol{\Phi}_c \in \mathbb{R}^{|\mathcal{E}_c| \times n}$ encodes its connectivity and is defined as

$$\phi_{c,li} = \begin{cases} 1 & \text{If node } i \text{ is the head of edge } l, \\ -1 & \text{If node } i \text{ is the tail of edge } l, \\ 0 & \text{otherwise.} \end{cases}$$

The row edge-incidence matrix $\boldsymbol{\Phi}_r \in \mathbb{R}^{|\mathcal{E}_r| \times m}$ is defined similarly. Assume that $\Theta$ non-empty, i.e. at least one entry of the matrix has been observed. Finally, assume that $\Omega$ is also coercive.

Note that any sequence $\mathbf{U}_t = a_t\mathbf{1}\mathbf{1}^\mathsf{T} + \mathbf{B}_t$ where $\langle \mathbf{B}_t, \mathbf{1}\mathbf{1}^\mathsf{T} \rangle = 0$. Note that $J_r(\mathbf{U}_t) = J_r(\mathbf{B}_t)$ and $J_c(\mathbf{U}_t) = J_c(\mathbf{B}_t)$. Let $\mathbf{U}_t$ be a diverging sequence, i.e. $\|\mathbf{U}_t\|_F \to \infty$. There are two cases to consider.

**Case I:** Suppose that $\|\mathbf{B}_t\|_F \to \infty$. Let

$$\mathbf{L} = \begin{pmatrix} \mathbf{I} \otimes \boldsymbol{\Phi}_r \\ \boldsymbol{\Phi}_c \otimes \mathbf{I} \end{pmatrix} \in \mathbb{R}^{|\mathcal{E}_r|m + |\mathcal{E}_c|n \times mn},$$

and let $\sigma_{\min}$ denote the smallest singular value of $\mathbf{L}$. Note that the null space of $\mathbf{L}$ is the span of $\mathbf{1}$. Therefore, since $\langle \mathbf{1}, \mathbf{b}_t \rangle = 0$

$$\|\mathbf{L}\mathbf{b}_t\|_2 \geq \sigma_{\min}\|\mathbf{B}_t\|_F. \qquad (13)$$

Also note that

$$\mathbf{L}\mathbf{b}_t = \begin{pmatrix} \mathrm{vec}(\boldsymbol{\Phi}_r\mathbf{B}_t) \\ \mathrm{vec}(\mathbf{B}_t\boldsymbol{\Phi}_c^\mathsf{T}) \end{pmatrix}.$$

Since the mapping $\mathbf{x} = \begin{pmatrix} \mathbf{x}_1^\mathsf{T} & \mathbf{x}_2^\mathsf{T} \end{pmatrix}^\mathsf{T} \mapsto \max\{\|\mathbf{x}_1\|_2, \|\mathbf{x}_2\|_2\}$ is a norm, and all finite dimensional norms are equivalent, there exists some $\eta > 0$ such that

$$\eta\|\mathbf{L}\mathbf{b}_t\|_2 \leq \max\left\{\|\boldsymbol{\Phi}_r\mathbf{B}_t\|_F, \|\mathbf{B}_t\boldsymbol{\Phi}_c^\mathsf{T}\|_F\right\}. \qquad (14)$$

By the triangle inequality

$$\max\left\{\|\boldsymbol{\Phi}_r\mathbf{B}_t\|_F, \|\mathbf{B}_t\boldsymbol{\Phi}_c^\mathsf{T}\|_F\right\} \leq \max\left\{\sum_{(i,j) \in \mathcal{E}_r} \|\mathbf{L}_{ij}\mathbf{b}_t\|_2, \sum_{(i,j) \in \mathcal{E}_c} \|\tilde{\mathbf{L}}_{ij}\mathbf{b}_t\|_2\right\}. \qquad (15)$$

Let $M = \max\{|\mathcal{E}_r|, |\mathcal{E}_c|\}$ then

$$\max\left\{\sum_{(i,j) \in \mathcal{E}_r} \|\mathbf{L}_{ij}\mathbf{b}_t\|_2, \sum_{(i,j) \in \mathcal{E}_c} \|\tilde{\mathbf{L}}_{ij}\mathbf{b}_t\|_2\right\} \leq M\max\left\{\max_{(i,j) \in \mathcal{E}_r} \|\mathbf{L}_{ij}\mathbf{b}_t\|_2, \max_{(i,j) \in \mathcal{E}_c} \|\tilde{\mathbf{L}}_{ij}\mathbf{b}_t\|_2\right\}. \qquad (16)$$

Putting inequalities (13), (14), (15), and (16) together gives us

$$\frac{\eta \sigma_{\min}}{M} \|\mathbf{B}_t\|_{\mathrm{F}} \leq \max \left\{ \max_{(i,j)\in\mathcal{E}_r} \|\mathbf{L}_{ij}\mathbf{b}_t\|_2, \max_{(i,j)\in\mathcal{E}_c} \|\tilde{\mathbf{L}}_{ij}\mathbf{b}_t\|_2 \right\}. \tag{17}$$

Since $\Omega$ is increasing according to Assumption 2.2, it follows that

$$\Omega\left(\frac{\eta\sigma_{\min}}{M}\|\mathbf{B}_t\|_{\mathrm{F}}\right) \leq \max\left\{\Omega\left(\max_{(i,j)\in\mathcal{E}_r}\|\mathbf{L}_{ij}\mathbf{b}_t\|_2\right), \Omega\left(\max_{(i,j)\in\mathcal{E}_c}\|\tilde{\mathbf{L}}_{ij}\mathbf{b}_t\|_2\right)\right\}. \tag{18}$$

Inequality (18) implies that

$$\begin{aligned}
\min\{\gamma_r,\gamma_c\}M\Omega\left(\frac{\eta\sigma_{\min}}{M}\|\mathbf{B}_t\|_{\mathrm{F}}\right) &\leq \min\{\gamma_r,\gamma_c\}\max\left\{J_r(\mathbf{U}_t), J_c(\mathbf{U}_t)\right\} \\
&\leq \gamma_r J_r(\mathbf{U}_t) + \gamma_c J_c(\mathbf{U}_t).
\end{aligned}$$

Consequently, since $\Omega$ is increasing and $\|\mathbf{B}_t\|_{\mathrm{F}} \to \infty$ implies that $f(\mathbf{U}_t) \to \infty$.

**Case II:** Suppose $\|\mathbf{B}_t\|_{\mathrm{F}} \leq B$ for some $B$. Then $|a_t| \to \infty$. Note that we have the following inequality

$$\begin{aligned}
f(\mathbf{U}_t) &\geq \sum_{(i,j)\in\Theta} (x_{ij} - b_{k,ij} - a_t)^2 \\
&\geq \sum_{(i,j)\in\Theta} a_t^2 - 2a_t(x_{ij} - b_{k,ij}) \\
&= |\Theta|a_t^2 - 2a_t \sum_{(i,j)\in\Theta}(x_{ij} - b_{k,ij}) \\
&\geq |\Theta|a_t^2 - 2a_t \sup_{\|\mathbf{B}_t\|_{\mathrm{F}}\leq B} \sum_{(i,j)\in\Theta}(x_{ij} - b_{k,ij}) \\
&= |\Theta|\left[a_t^2 - 2a_t C\right] \\
&= |\Theta|\left[(a_t - C)^2 - C^2\right],
\end{aligned}$$

where $C = |\Theta|^{-1} \sup_{\|\mathbf{B}_t\|_{\mathrm{F}}\leq B} \sum_{(i,j)\in\Theta}(x_{ij} - b_{k,ij})$.

The function $(a_t - C)^2$ diverges since $|a_t| \to \infty$. Therefore, the function $f$ is coercive.

## C FILLING IN MISSING DATA

We present the original underlying structure of 3D points used to generate the Euclidean distance matrix $\mathbf{X}$ for the datasets **linkage** and **linkage2** in Figure 5 and Figure 8. In Figure 6 and Figure 9, on the left we plot the original complete matrix where the rows and columns have been ordered according to the geometry of the 3D points. On the right we plot the matrix we analyze whose rows and columns have been permuted and $50\%$ of the entries have been removed. In Figure 7 and Figure 10 we display the matrix $\tilde{\mathbf{X}}^{(l,k)}$ for three pairs of values $l, k$ to demonstrate the smoothing that is occurring across the different scales of the rows and columns.

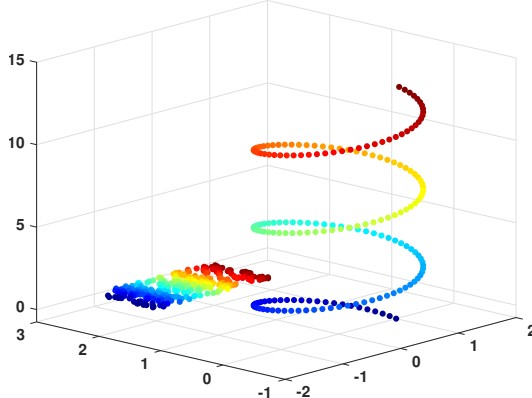

Figure 5: Points in 3D used to generate the Euclidean distance matrix $\mathbf{X}$ in the **linkage** dataset. Rows correspond to the helix, columns to the 2D surface. Points are colored corresponding to the embedding of rows and columns in Figure 2.

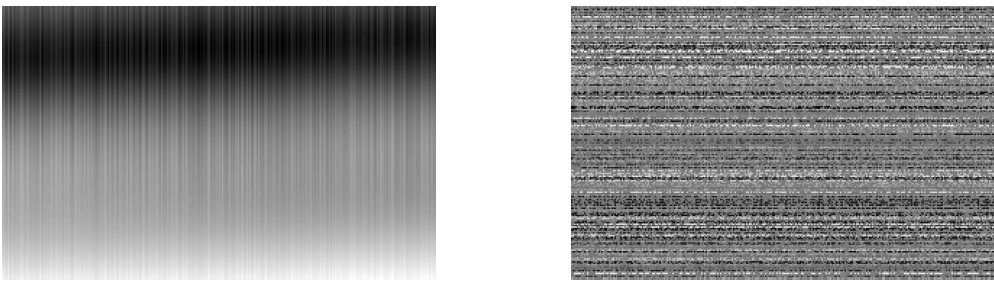

Figure 6: **linkage** dataset: (Left) Complete matrix $\mathbf{X}$. (Right) Matrix whose rows and columns and columns have been permuted and $50\%$ of the values have been removed.

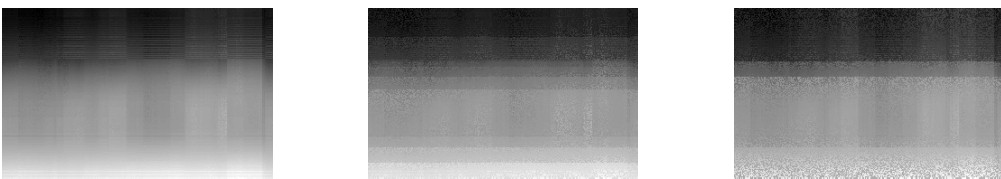

Figure 7: **linkage** dataset: Filled-in matrices $\tilde{\mathbf{X}}$ at multiple scales: $\tilde{\mathbf{X}}^{(-3,-2)}, \tilde{\mathbf{X}}^{(1,0)}, \tilde{\mathbf{X}}^{(5,2)}$. Rows and columns have been reordered based on the manifold embedding following (Ankenman, 2014).

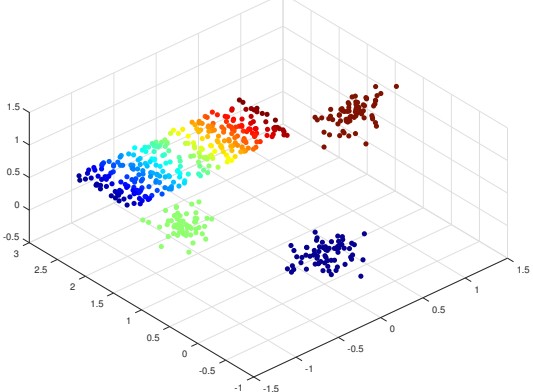

Figure 8: Points in 3D used to generate the Euclidean distance $\mathbf{X}$ in the **linkage2** dataset. Rows correspond to the three 3D Gaussians, columns to the 2D surface. Points are colored corresponding to the embedding of rows and columns in Figure 2

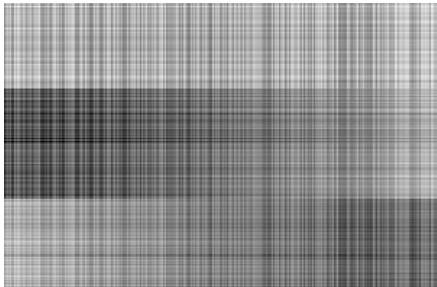 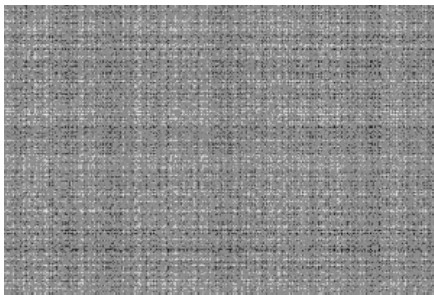

Figure 9: **linkage2** dataset: (Left) Complete matrix $\mathbf{X}$. (Right) Matrix whose rows and columns and columns have been permuted and $50\%$ of the values have been removed.

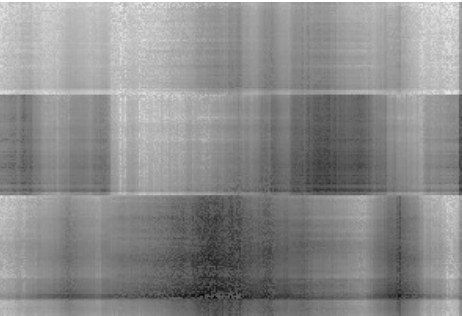

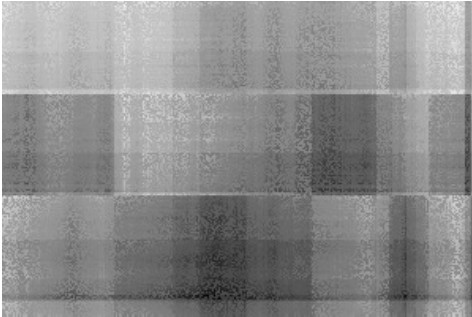

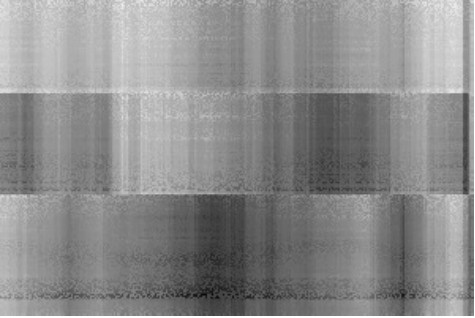

Figure 10: **linkage2** dataset: Filled-in matrices $\tilde{\mathbf{X}}$ at multiple scales: $\tilde{\mathbf{X}}^{(-4,-3)}, \tilde{\mathbf{X}}^{(-1,1)}, \tilde{\mathbf{X}}^{(5,-3)}$. Rows and columns have been reordered based on the manifold embedding following (Ankenman, 2014).

