# OpenReview forum: "Co-manifold learning with missing data"
_ICLR.cc/2019/Conference_

### Official Review · AnonReviewer1 · 2018-11-01
**Sufficient Novelty but Significant Clarity Issues**

**Rating:** 7
**Confidence:** 4

**Review:**

Review for CO-MANIFOLD LEARNING WITH MISSING DATA
Summary:
This paper proposes a two-stage method to recovering the underlying structure of a data manifold using both the rows and columns of an incomplete data matrix. In the first stage they impute the missing values using their proposed co-clustering algorithm and in the second stage they propose a new metric for dimension reduction.
The overall motivation for how they construct the algorithm and the intuition behind how all the pieces of the algorithm work together are not great. The paper also has significant specific clarity issues (listed below). Currently these issues seem to imply the proposed algorithm has significant logic issues (mainly on the convex/concave confusions); however depending on how they are addressed, this may end up not being an issue. The experimental results for the two simulated datasets look very good. However for the lung dataset, the results are less promising and it is less clear of the advantage of the proposed algorithm to the two competing ones.
Novelty/Significance:
The overall idea of the algorithm is sufficiently novel. It is very interesting to consider both rows and column correlations. Each piece of the algorithm seems to draw heavily on previous work; bi-clustering, diffusion maps, but overall the idea is novel enough. The algorithm is significant in that it addresses a relatively open problem that currently doesn’t have a well established solution.
Questions/Clarity:
Smooth is not clearly defined and not an obvious measure for a matrix. Figure 1 shows smooth matrices at various levels, but still doesn’t define explicitly what smoothness is. Does smoothness imply all entries are closer to the same value?
 “Replacing Jr(U) and Jc(U) by quadratic row and column Laplacian penalties” – The sentence is kind of strange as Laplacian penalties is not a thing. Graph Laplacian can be used as an empirical estimate for the Laplace Beltrami operator which gives a measure of smoothness in terms of divergence of the gradient of a function on a manifold; however the penalty is one on a function’s complexity in the intrinsic geometry of a manifold. It is not clear how the proposed penalty is an estimator for the intrinsic geometry penalty. It seems like the equation that is listed is just the function map Omega(x) = x^2, which also is not a concave function (it is convex), so it does not fit the requirements of Assumption 2.2.
Proposition 1 is kind of strangely presented. At first glance, it is not clear where the proof is, and it takes some looking to figure out it is Appendix B because it is reference before, not after the proposition. Or it might be more helpful if it is clearly stated at the beginning of Appendix B that this is the proof for Proposition 1.
The authors write: “Missing values can sabotage efforts to learn the low dimensional manifold underlying the data. … As the number of missing entries grows, the distances between points are increasingly distorted, resulting in poor representation of the data in the low-dimensional space.” However, they use the observed values to build the knn graph used for the row/column penalties, which is counter-intuitive because this knn graph is essentially estimating a property of a manifold and the distances have the same distortion issue.
Why do the author’s want Omega to be concave functions as this makes the objective not convex. Additionally the penalty sqrt(|| ||_2) is approximately doing a square root twice because the l2-norm already is the square root of the sum of squares. Also what is the point of approximating the square root function instead of just using the square root function? It is overall not clear what the nature of the penalty term g2 is; Appendix A, implies it must be overall a convex function because of the upper bound.
Equation 5 is not clear that it is the first order taylor approximation. Omega’ is the derivative of the Omega function? Do the other terms cancel out? Also what is the derivative with respect to; each Ui. for all Uj. ?
 “first-order Taylor approximation of a differentiable concave function provides a tight bound on the function” – Tight bound is not an appropriate term and requires being provable. Unless the function is close to linear, a first order Taylor approximation won’t be anything close to tight.
The authors state the objective in 1 is not convex. Do they mean it is not strictly convex? In which case, by stationary points, they are specifically referring to local minima? Otherwise, what benefits does the MM algorithm have on an indefinite objective i.e. couldn’t you end up converging to a saddle point or a local maxima instead of a local minima, as these are all fixed points.
It is not clear what the sub/super scripts l, k mean. Maybe with these defined, the proposed multi-scale metric would have obvious advantages, but currently it is not clear what the point of this metric is.
Figure 4 appears before it is mentioned and is displayed as part of the previous section.
For the Lung data, it does not look like the proposed algorithm is better than the other two. None of the algorithms seem to do great at capturing any of the underlying structure, especially in the rows. It also is not super clear that the normal patients are significantly further from the cancer patients. Additionally are the linkage results from figure 3 from one trial? Without multiple trials it is hard to argue that this not just trial noise.
How big are N1 and N2 in the linkage simulations. The Lung dataset is not very large, and it seems like the proposed algorithm has large computation complexity (it is not clear). Will the algorithm work on even medium-large sized matrices (10^4 x 10^4)?

---

> ### Author Response · Authors · 2018-11-27
> **response to AnonReviewer1 (part 1)**
>
> 1. The overall motivation for how they construct the algorithm and the intuition behind how all the pieces of the algorithm work together are not great.
>
> A. Corrected, we have provided more motivation intuition and details on the algorithm.
>
> 2. Smooth is not clearly defined and not an obvious measure for a matrix. Figure 1 shows smooth matrices at various levels, but still doesn't define explicitly what smoothness is. Does smoothness imply all entries are closer to the same value?
>
> A. Smoothness can be characterized mathematically using a bi-Hölder condition, which is a common assumption in the matrix organization / biclustering literature. Smoothness implies that under the true row and column geometry of the data, neighboring entries are similar.
>
> 3. "Replacing Jr(U) and Jc(U) by quadratic row and column Laplacian penalties" The sentence is kind of strange as Laplacian penalties is not a thing. Graph Laplacian can be used as an empirical estimate for the Laplace Beltrami operator which gives a measure of smoothness in terms of divergence of the gradient of a function on a manifold; however the penalty is one on a function's complexity in the intrinsic geometry of a manifold. It is not clear how the proposed penalty is an estimator for the intrinsic geometry penalty. It seems like the equation that is listed is just the function map $\Omega(x) = x^2$, which also is not a concave function (it is convex), so it does not fit the requirements of Assumption 2.2.
>
> A. Corrected. We have removed the term "Laplacian penalties."  The function $\Omega(x) = x^2$ is convex, but it is not the $\Omega$ function studied in this paper. We included it as part of the literature review as it is a commonly used regularizer that bears some similarity to the one used in this paper, but agree we did not make this clear enough. We have added discussion below Assumption 2.2, clarifying that the penalties used in this paper (ones that satisfy Assumption 2.2) are different from those like commonly used convex quadratic penalties, previously referred to as ``"Laplacian penalties." Penalties used in this paper can completely eliminate small variations between pairs of similar rows (columns) but less aggressively shrink very different pairs of rows (columns) towards each other.
>
> 4. Proposition 1 is kind of strangely presented. At first glance, it is not clear where the proof is, and it takes some looking to figure out it is Appendix B because it is reference before, not after the proposition. Or it might be more helpful if it is clearly stated at the beginning of Appendix B that this is the proof for Proposition 1.
>
> A. Corrected. We have put the sentence "The proof of Proposition 1 is in Appendix B" after the statement of the proposition. We have also changed the title of Appendix B from "Convergence" to "Proof of Proposition 1."
>
> 5. The authors write: "Missing values can sabotage efforts to learn the low dimensional manifold underlying the data. As the number of missing entries grows, the distances between points are increasingly distorted, resulting in poor representation of the data in the low-dimensional space." However, they use the observed values to build the knn graph used for the row/column penalties, which is counter-intuitive because this knn graph is essentially estimating a property of a manifold and the distances have the same distortion issue.
>
> A. While we are using a knn graph on the rows and columns, note that our method takes into account both rows and columns geometry jointly. Thus we are leveraging information from both domains to fill in the values of the data. In addition the weights of of our knn graph are continuously updated throughout the optimization based on the current smooth estimate $U$. Thus the weights are pulling rows and columns together at increasingly coarse scales, going from local geometry to global geometry.

---

> > ### Author Response · Authors · 2018-11-27
> > **response to AnonReviewer1 (part 2)**
> >
> > 6. Why do the authors want Omega to be concave functions as this makes the objective not convex. Additionally the penalty $\sqrt(|| \cdot ||_2) $is approximately doing a square root twice because the l2-norm already is the square root of the sum of squares. Also what is the point of approximating the square root function instead of just using the square root function? It is overall not clear what the nature of the penalty term g2 is; Appendix A, implies it must be overall a convex function because of the upper bound.
> >
> > A. Corrected. In a new paragraph above Section 2.1, we explain why we choose to solve a non-convex optimization problem compared to a convex one. Using a single square-root would result in a convex $\Omega$. We use a function that results in taking a square root twice to  get a concave $\Omega$. We approximate the square root because just using a square root function, i.e. taking $\Omega(z) = \sqrt{z}$, would result in an $\Omega$ that does not satisfy Assumption 2.2 as the derivative $\Omega'(z) = \frac{1}{2\sqrt{z}}$ does not exist at $z=0$. Practically this would mean that the MM algorithm updates would be undefined if for example any pair of rows or column variables were identical (as desired), e.g. if $U_{i \cdot} = U_{j \cdot}$. The penalty term g2 in Appendix A is indeed convex. Our MM algorithm inexactly solves the non-convex optimization problem, whose objective function is given in Eq (1), by solving a sequence of convex optimization problems - the convex biclustering problem, whose objective function is given in the equation above Eq (5).
> >
> > 7. Equation 5 is not clear that it is the first order Taylor approximation. Omega' is the derivative of the Omega function? Do the other terms cancel out? Also what is the derivative with respect to; each Ui. for all Uj. ?
> >
> > A. Corrected. $\Omega'$ is the derivative of the $\Omega$ function. We have clarified where the notation is first introduced. The inequality $\Omega(z) \leq \Omega(\tilde{z}) + \Omega'(\tilde{z})(z - \tilde{z})$ holds for all non-negative $z$ and $\tilde{z}$. Therefore, it holds for $z = \lVert U_{i\cdot} - U_{j \cdot} \rVert_2$ and $\tilde{z} = \lVert \tilde{U}_{i\cdot} - \tilde{U}_{j \cdot} \rVert_2$.
> >
> > 8. "first-order Taylor approximation of a differentiable concave function provides a tight bound on the function" Tight bound is not an appropriate term and requires being provable. Unless the function is close to linear, a first order Taylor approximation won't be anything close to tight.
> >
> > A. We used "tight" in the sense that there are values of $z$ at which the inequality becomes an equality. Note that the linear approximation $\Omega(\tilde{z}) + \Omega'(\tilde{z}) (z - \tilde{z})$ equals the nonlinear function $\Omega(z)$ when $z = \tilde{z}$. This tangency condition is needed to prove the convergence properties of the MM algorithm. But we agree "tight" was not appropriate and have changed "tight bound"to "global upper bound."

---

> > > ### Author Response · Authors · 2018-11-27
> > > **response to AnonReviewer1 (part 3)**
> > >
> > > 9. The authors state the objective in 1 is not convex. Do they mean it is not strictly convex? In which case, by stationary points, they are specifically referring to local minima? Otherwise, what benefits does the MM algorithm have on an indefinite objective i.e. couldn't you end up converging to a saddle point or a local maxima instead of a local minima, as these are all fixed points.
> > >
> > > A. In a new paragraph on page 4, we have clarified that the objective is not convex when using $\Omega$ that are concave and more importantly we explain why we choose to solve a non-convex optimization problem compared to a convex one.
> > >
> > > Minimizing a non-convex function, however, is generically hard and converging to a stationary point is typically the best one can hope for when attempting to minimize a non-convex function (There are rare exceptions, notably using the SVD to find the best, in Frobenius-nom, rank-k approximation of a matrix). Thus, not being able to guarantee convergence to a global minimizer is not a drawback that is unique to the MM algorithm. One could employ other algorithms such as proximal gradient or ADMM to minimize the non-convex objective and at best would only be able to guarantee convergence to a stationary point of the original problem. The objective function landscapes of deep learning problems are quite non-convex, and yet the humble stochastic gradient method produces useful answers even though convergence to a global minimizer cannot be guaranteed.
> > >
> > > Moreover, it is not essential to use an MM algorithm, but we use it to take advantage of existing algorithms for the convex biclustering problem. As noted in the sentence after Eq 5: "Minimizing $g(U \mid \tilde{U} )$ is equivalent to minimizing the objective function of the convex biclustering problem for which efficient algorithms have been introduced (Chi et al., 2017)."
> > >
> > > Due to space limitation we have decided to keep the detailed discussion on stationary points in Appendix B.2 from the original submission, but to answer the referee's question, this is what we mean by a stationary point. A point $u$ is a stationary point of a function $f$ if all directional derivatives of $f$ at $u$ are non-negative:
> > > $$
> > > \underset{t \rightarrow 0}{\lim}\; \frac{f(u + tv) - f(u)}{t} \geq 0 \quad\quad \text{for all $v$ such that $u + tv$ is in the domain of $f$}.
> > > $$
> > > In other words, taking an infinitesimal step in any direction $v$ cannot decrease the objective function value but is allowed to increase the objective function. Note that the directional derivative of the function in Eq (1) exists everywhere and for all directions $v$.
> > >
> > > 10. It is not clear what the sub/super scripts $l, k$ mean. Maybe with these defined, the proposed multi-scale metric would have obvious advantages, but currently it is not clear what the point of this metric is.
> > >
> > > A.  Corrected. We now explicitly write that "Note that the $l$ and $k$ denote the power of 2 taken for specific row and column cost parameters ($\gamma_c,\gamma_r$) in the solution. This is intended as a compact notation that corresponds a pair of parameters $(\gamma_r,\gamma_c)$ to their solution $U^{l,k}$ and filled in estimate $\tilde{X}^{(l,k)}$."
> > >
> > > We also write "Our goal is to aggregate distances between a pair of rows (columns) across multiple scales of the solution, to calculate a metric that better recovers the local and global geometry of the data despite the missing values, thus "fixing" the missing data metric."
> > > And furthermore "This metric takes advantage of solving the optimization for multiple pairs of cost parameters and filling in the missing values with increasingly smooth estimates (as $\gamma_r$ and $\gamma_c$ increase). It also alleviates the need to identify the ideal scale at which to fill in the points; it is not clear that a single "optimal" scale actually exists, but rather different points in the matrix may have different optimal scales". We have demonstrated in the experimental results the large variability in results for a competing method for which it is unclear how to select the proper scale of the row and column cost parameters.
> > >
> > > 11. Figure 4 appears before it is mentioned and is displayed as part of the previous section.
> > >
> > > A. Corrected.

---

> > > > ### Author Response · Authors · 2018-11-27
> > > > **response to AnonReviewer1 (part 4)**
> > > >
> > > > 12.  For the Lung data, it does not look like the proposed algorithm is better than the other two. None of the algorithms seem to do great at capturing any of the underlying structure, especially in the rows. It also is not super clear that the normal patients are significantly further from the cancer patients.
> > > >
> > > > A. Corrected. We have added clarification as to which color in the plot corresponds to which sample type. in addition we added experimental results for an additional method that jointly takes into account graph structure on the rows and columns. This method similar to ours separates between the normal and the colon cancer patients.
> > > >
> > > > With respect to the organization of the rows, the manifold-like structure revealed by all methods indeed serves to illustrate our motivation for this paper. Our assumption is that for certain types of data, assuming a bi-clustering model is too restrictive and results in breaking up smooth geometries into disjoint clusters that do not match the actual geometry of the data. For the genes, the structure implied by the data is not one of disjoint clusters. (We have witnessed this manifold structure of genes in other gene expression datasets and indeed this one of of the motivations for our approach).
> > > >
> > > > 13. Additionally are the linkage results from figure 3 from one trial? Without multiple trials it is hard to argue that this not just trial noise.
> > > >
> > > > A. Corrected. We now write in the text "where we averaged over 30 realizations of the data and the locations of the missing entries".
> > > >
> > > > 14. How big are N1 and N2 in the linkage simulations. The Lung dataset is not very large, and it seems like the proposed algorithm has large computation complexity (it is not clear). Will the algorithm work on even medium-large sized matrices ($10^4 x 10^4$)?
> > > >
> > > > A. Corrected. We now write in the text for linkage1 that $N1=190, N2=300$ and for linkage 2 $N1=200, N2=300$.
> > > > Regarding computational complexity, we now write in the conclusions "We also intend to develop efficient solutions to accelerate the optimization in order to address large-scale datasets, in addition to the small-scale regime we demonstrate here. We note that the datasets  considered here, while being small-scale in the observation domain are high-dimensional in the feature domain, which is a non-trivial setting, and indeed a challenge for supervised methods such as deep learning due to limited training data."

---

### Official Review · AnonReviewer3 · 2018-11-02
**A joint method for filling missing values and emdedding rows & columns, but not convincing**

**Rating:** 4
**Confidence:** 3

**Review:**

This paper presents a joint learning method for filling missing value and bi-clustering. The method extends (Chi et al. 2017), using a penalized matrix approximation. The proposed method is tested on three data sets, where two are synthetic and one small real-world data matrix. The presented method is claimed to be better than two classical approaches Nonlinear PCA and Diffusion Maps.

1) Filling missing values is not new. Even co-clustering with missing values also exists. It is insufficient to defeat two methods which are older than ten years. More extensive comparison is needed but lacking here. Why not first use a dedicated method such as MICE or collaborative filtering, and then run embedding method on rows and columns?

2) The purpose of the learning is unclear. The title does not give any hint about the learning goal. The objective function reads like filling missing values. The subsequent text claims that minimizing such a objective can achieve biclustering. However, in the experiment, the comparison is done via visualization and normal clustering (k-means).

3) The empirical results are not convincing. Two data sets are synthetic. The only real-world data set is very small. Why k-means was used? How to choose k in k-means?

4) The choice Omega function after Proposition 1 needs to be elaborated. A function curve plot could also help.

5) What is Omega' in Eq. 2?

---

> ### Author Response · Authors · 2018-11-27
> **response to AnonReviewer3 (Part 1/2)**
>
> 1a) Filling missing values is not new. Even co-clustering with missing values also exists.
>
> A. Corrected.
> Note that our end-goal is not to fill in the missing data, but rather to reveal low dimensional embeddings for rows and columns of a data matrix in a missing data scenario.
> In addition, we are not addressing a co-clustering scenario but rather a more general problem in which the rows and columns are not necessarily clustered, but rather lie on a manifold structure.
> Taking the reviewers comments into consideration we have updated the introduction, now writing that
> "...In certain settings, however, assuming a bi-clustering model is too restrictive and results in breaking up smooth geometries into artificial disjoint clusters that do not match the actual structure of the data. This can occur when the true geometry is one of overlapping rather than disjoint clusters, for example in word-document analysis (Ahn et al., 2010), or when the underlying structure is not one of clusters at all but rather a smooth manifold (Gavish & Coifman, 2012). Thus, we consider a more general viewpoint: data matrices possess geometric relationships between their rows (features) and columns (observations) such that both modes lie on low-dimensional manifolds."
>
> We have also added more details on how our formulation differs from related work, now writing that
> "Our formulation (1) is distinct from related problem formulations in the following ways:
> 1. Rows and columns of U are simultaneously shrunk towards each other as the parameters $\gamma_r$ and $\gamma_c$ increase. Note that this shrinkage procedure is fundamentally different from methods like the clustered dendrogram, which independently cluster the rows and columns as well as alternating partition tree construction procedures (Gavish & Coifman, 2012; Mishne et al., 2016).
> 2. Our ultimate goal is not to perform matrix completion (though this is a by-product of our approach) but rather to perform joint row and column dimension reduction.
> 3. Our work generalizes both  Shahid et al. (2016) and Chi et al. (2017) in that we seek the flexibility of performing non-linear dimension reduction on the rows and columns of the data matrix, i.e. a more general manifold organization than a co-clustered structure.
> 4. Instead of determining an optimal single scale of the solution as in Shahid et al.(2016);Chi et al. (2017), we recognize that the multiple scales of the different solutions can be aggregated to better estimate the underlying geometry, similar to the tree-based Earth mover's distance proposed in Ankenman (2014); Mishne et al. (2017)."
>
> 1b) It is insufficient to defeat two methods which are older than ten years. More extensive comparison is needed but lacking here. Why not first use a dedicated method such as MICE or collaborative filtering, and then run embedding method on rows and columns?
>
> A. Corrected. We have added both qualitative and quantitative comparisons to Fast robust PCA on Graphs [Shahid2016] in the experimental results.
> We note that we tried to impute the data using MICE as the reviewer suggested but the algorithm failed to converge in reasonable time on either dataset. From our understanding MICE is also unsuitable for filling in the data for high percentage of missing values as we considered in our paper.
>
> Comparison to Diffusion Maps is intended to demonstrate the degradation that occurs for both visualization and clustering of data when values are missing. Since we use diffusion maps in our framework, this is a natural comparison. In addition, comparing to Diffusion Maps when the values have  been filled in with the mean of all data is s equivalent to applying our approach for only a single scale of the cost parameters: $\gamma_r,\gamma_c \rightarrow \infty$.

---

> > ### Author Response · Authors · 2018-11-27
> > **response to AnonReviewer3 (Part 2/2)**
> >
> > 2) The purpose of the learning is unclear. The title does not give any hint about the learning goal. The objective function reads like filling missing values. The subsequent text claims that minimizing such a objective can achieve biclustering. However, in the experiment, the comparison is done via visualization and normal clustering (k-means).
> >
> > A.  Manifold learning is a class of unsupervised methods aiming at uncovering a low-dimensional representation for data. We have generalized the typical manifold learning problem to addressing a coupled structure along both rows and columns, revealing manifolds for those. The purposes for such learning, as we present in the introduction, are plentiful and include exploratory data analysis, data visualization, and precursors for clustering and classification.
> >
> > Regarding biclustering, following our reply above to 1a),  we integrate multiple biclustering solutions in order to perform co-manifold learning, i.e. simultaneously identify row and column space manifolds organizing the entries of the data matrix, as opposed to seeking a single bi-clustering. Our assumption is that for different types of data, assuming a bi-clustering model is too restrictive and results in breaking up smooth geometries into disjoint clusters that do not match the actual geometry of the data.
> >
> > Our empirical results are intended to address these cases, demonstrated for both gene expression and a synthetic example. In both cases the domain of the columns is clustered whereas the domain of the rows is a manifold. Indeed the visualization of the low-dimensional embedding of the genes for all methods demonstrates that there is no clear clustered structure in this domain (we have witnessed this manifold structure of genes in other gene expression datasets and indeed this one of of the motivations for our approach).
> > Following the reviewer's comment we now clarify in the beginning of the experimental results section that
> > "The model we consider in the paper is such that the data is not represented by a biclustering model but rather at least one of the modes (rows/columns) lies on a low-dimensional manifold. In our experiments we consider three such examples. In the first a manifold structure exists along both rows and columns, and for the second and third the columns belong to disjoint clusters while the rows lie on a manifold..."
> >
> > 3) The empirical results are not convincing. Two data sets are synthetic. The only real-world data set is very small. Why k-means was used? How to choose k in k-means?
> >
> > A. Corrected.
> >  k-means was used as it a common technique to extract clusters from low-dimensional embeddings (e.g. spectral clustering).
> >  We now write that "We apply $k$-means to the column embeddings of each method, with $k$ set to the correct number of clusters in the data, as we want to evaluate the ability of the methods to properly represent the data without being sensitive to empirical estimation of the number of clusters in the data."
> > Regarding the size of the datasets, we now write in the conclusions:
> > "We also intend to develop efficient solutions to accelerate the optimization in order to address large-scale datasets, in addition to the small-scale regime we demonstrate here. We note that the datasets  considered here, while being small-scale in the observation domain are high-dimensional in the feature domain, which is a non-trivial setting, and indeed a challenge for supervised methods such as deep learning due to limited training data."
> >
> > 4. The choice Omega function after Proposition 1 needs to be elaborated. A function curve plot could also help.
> >
> > A. Corrected. We have added discussion below Assumption 2.2 to clarify the key feature of the choice of the $\Omega$ function (previously after Proposition 1, but now in Eq. 3). We have also added a function curve plot in Appendix A when discussing the construction of the majorization.
> >
> > 5. What is Omega' in Eq. 2?
> >
> > A. Corrected. $\Omega'(z)$ is the first derivative of the function $\Omega(z)$. We have added an explanation of this where the notation is first introduced.

---

### Official Review · AnonReviewer2 · 2018-11-03
**lack of novelty**

**Rating:** 4
**Confidence:** 4

**Review:**

The manuscript proposes a co-manifold learning approach for missing data.  The problem is important, but the method is lack of novelty.
Pros: important problem setting, Good experimental results.
Cons: the method is lack of novelty.

In detail, the method just simply combines a loss for competing missing values, which is not new,  and Laplacian losses for rows and columns, which are also not new. I don't see much novelty in the model.

---

> ### Author Response · Authors · 2018-11-27
> **response to AnonReviewer2**
>
> Q. In detail, the method just simply combines a loss for competing missing values, which is not new,  and Laplacian losses for rows and columns, which are also not new.
>
> A. Corrected. We have added a clarification on how our work is different from related work. Specifically, we clarify below Assumption 2.2, that our penalties are different from commonly used Laplacian losses and the advantage our penalties have over commonly used Laplacian penalties for rows and columns.
> We also now write that:
>
> "Our formulation (1) is distinct from related problem formulations in the following ways:
> 1. Rows and columns of U are simultaneously shrunk towards each other as the parameters $\gamma_r$ and $\gamma_c$ increase. Note that this shrinkage procedure is fundamentally different from methods like the clustered dendrogram, which independently cluster the rows and columns as well as alternating partition tree construction procedures (Gavish & Coifman, 2012; Mishne et al., 2016).
> 2. Our ultimate goal is not to perform matrix completion (though this is a by-product of our approach) but rather to perform joint row and column dimension reduction.
> 3. Our work generalizes both  Shahid et al. (2016) and Chi et al. (2017) in that we seek the flexibility of performing non-linear dimension reduction on the rows and columns of the data matrix, i.e. a more general manifold organization than a co-clustered structure.
> 4. Instead of determining an optimal single scale of the solution as in Shahid et al.(2016);Chi et al. (2017), we recognize that the multiple scales of the different solutions can be aggregated to better estimate the underlying geometry, similar to the tree-based Earth mover's distance proposed in Ankenman (2014); Mishne et al. (2017)."

---

### Author Response · Authors · 2018-11-27
**Revised paper**

We thank the reviewers for their time and constructive feedback. We have uploaded a revised version of the paper.
We have added comparisons with an additional recently published method in the experimental results section, and have provided more details on the motivation to our approach, its novelty with respect to related work and the choice of row and column penalties used in our problem formulation. Detailed individual replies to reviewers are added below.

---

### Meta-Review · Area_Chair1 · 2018-12-14

**Confidence:** 4
**Recommendation:** Reject

**Metareview:**

This manuscript proposes a technique for co-manifold learning that exploits smoothness jointly over the rows and columns of the data. This is an important topic worth further study in the community.

The reviewers and AC opinions were mixed, with reviewers either being unconvinced about the novelty of the proposed work or expressing issues about the clarity of the presentation. Further improvement of the clarity -- particularly clarification of the learning goals, combined with additional convincing experiments would significantly strengthen this submission.